# Trends in the erythemal radiant exposure from re-evaluated measurements (1976−2023) with biometers at Belsk, Poland, and their sources from corresponding ozone, aerosol, and cloud observations

Agnieszka Czerwińska[1], Janusz Krzyścin[1], Janusz Jarosławski[1], Piotr S. Sobolewski[1], Aleksander Pietruczuk[1]

[1]Institute of Geophysics, Polish Academy of Sciences, Warsaw, 01-452, Poland

*Correspondence to*: Agnieszka Czerwińska (aczerwinska@igf.edu.pl)

**Abstract.** The world's longest homogenised series of measured erythemal solar irradiance comes from biometers operating at Belsk, Poland (51.84° N, 20.79° E) in 1976−2023. Linear trends in erythemal radiant exposure (ERE) are calculated for the first (1976−1999) and second (2000−2023) halves of the observation period. A statistically significant increasing trend of 6.9% per decade was found for annual ERE in the first period. In the second period, only the trend in seasonal (June-August) ERE was statistically significant (3.1% per decade). The method proposed here to reveal the sources of the ERE trends involves the construction of separate and combined forcings from clear sky (total column ozone and aerosol optical depth) and cloud proxies (sunshine duration, clearness index). The superposition of these proxy effects over 1985−1999 was the source of the positive trend in annual ERE for the first half of the observations. Before 1985, clear sky and cloud effects had cancelled each other. The maximum ERE growth rate of 19.4% per decade over 10 years was found in 1984−1993, with overlapping forcing effects of decreasing ozone and cloudiness responsible. Clear sky and cloud effects stabilised after 1996 and 2005, respectively. The cloud effect has begun to force a positive trend in the annual ERE again since 2015 due to increasing cloud transparency and/or the disappearance of cloud cover. Comparisons of the performance of linear and non-linear versions of the ERE models show that interaction effects between clear sky and cloud proxies can be neglected in trend analyses.

## 1 Introduction

UV radiation (UVR) passing through the atmosphere is attenuated because of absorption and scattering by gaseous substances (e.g. absorption mainly by $O_3$, $NO_2$ and $SO_2$ in the UV-B region of the solar spectrum, and Rayleigh scattering by atmospheric molecules, with the strongest influence from the most abundant nitrogen and oxygen molecules), aerosols (e.g. absorption by soot particles and Mie scattering by dust particles), and cloud particles (scattering by cloud droplets and ice crystals). The path length of solar radiation to the ground depends on the sun's elevation and the altitude of the site, so the time of day, the day of the year and the geographical coordinates are also important factors in surface UV variability on intraday and seasonal time scales (DeLuisi, 1997).

Studies of changes in surface UVR have usually been combined with analyses of corresponding changes in its forcing factors, such as ozone, aerosols and clouds (e.g. Calbó et al., 2005; Borkowski, 2008; Krzyścin et al., 2011; Fountoulakis et al., 2018; Aun et al., 2019; Bernhard et al., 2023). Excessive UV radiation has been linked to a number of health problems (Neale et al., 2023), such as erythema appearance, skin cancer (including the deadly melanoma), cataracts and immunosuppression (Young, 2006, Bernard et al., 2019). On the other hand, UVR-induced cutaneous synthesis is the main source of vitamin $D_3$ in the human body (Holick and Chen, 2008). In additions to being the main source of vitamin D, exposure to UVR has other positive effects, e.g., lowering the blood pressure, psoriasis clearing, improving mood by endorphin release, and increases the melanin production in skin (Juzeniene and Moan, 2012; Trummer et al., 2016). Since the mid-1980s, there has been interest in changes in erythemal solar radiation at ground level, along with the observed downward trend in stratospheric ozone over Antarctica.

This tendency appeared in the early 1980s as a result of the accumulation of man-made ozone-depleting substances (chlorofluorocarbons) in the stratosphere (Molina and Rowland, 1974; Farman et al.,1985; Solomon et al., 1986).

Erythemal UV irradiance is the intensity of UV radiation that causes sunburn, i.e., UV spectrum weighted with action spectrum of erythema (sunburn) appearance and integrated over the wavelength range from 280 to 400 nm. The commonly used 1 UV Index is equivalent to 25 mW m$^{-2}$ of erythemal UV irradiance (WHO, 2002).

A global erythemal irradiance observing network of about 15 stations, was established in the early 1970s (WMO, 1977). This followed the construction of the Robertson-Berger (RB) meter, which made it possible to monitor erythemal irradiance (Berger,

1976), rather than the recognition of the urgent need for such observations. RB meter was also installed in 1975 at the Central Geophysical Observatory of the Institute of Geophysics (IG), Polish Academy of Sciences (PAS), located at Belsk, i.e., a rural site in central Poland. Regular monitoring has been started here in January 1976 as part of the geophysical monitoring including also parameters important to its variability as total column ozone (TCO$_3$) by the Dobson spectrophotometer, and cloud attenuation of solar radiation by Cambell-Stokes sunshine recorder and global solar irradiance by various pyranometers. Since

the early 1990s, other next-generation broadband biometers were used for UV monitoring, including the Solar Light (SL) Model 501 A biometer (#927 and #2011 for the period 1993−1994 and 1995−2013, respectively) and the Kipp & Zonen (KZ) UV-AE-T #30616 from 5 August 2013 to the present. The homogenisation of the 1976−2013 SL time series was performed by comparing the measured erythemal radiant exposure (ERE) under clear skies with the output of the Tropospheric Ultraviolet and Visible (TUV) Radiation Transfer Model (TUV, 2025). The quality of the KZ measurements (2013−2023) was supported

by the well-maintained Brewer Mark II #64 spectrophotometer. The SHICRivm algorithm is used for quality checks and to extend the Brewer spectra to 400 nm (Slaper et al, 1995). To retrieve daily erythemal radiant exposures for comparison with KZ, spectra are weighted with the CIE erythemal action spectrum (CIE 2019) and integrated over wavelengths (from 280 to 400 nm) and time (sunrise to sunset). Time series of UV ground-based measurements longer than four decades are very rare. They are not erythemally weighted (Chubarova et al., 2000) or do not come from biometers, but rather spectral measurements

(NDACC, 2025, WOUDC, 2025). Therefore, it can be stated that the world's longest homogenised time series of measured erythema-weighted solar irradiance comes from biometers operating at Belsk (Krzyścin et al., 2025).

This article focuses on the investigation of the long-term variability of monthly, seasonal, and annual ERE and its sources, taking into account the co-observed time series of factors influencing UVR, i.e. TCO$_3$, aerosol optical depth (AOD) and different characteristics (sunshine duration, global solar irradiance, clearness index) of solar radiation attenuation by clouds.

The paper is organised as follows. The data used are described in Sect. 2. The trend analyses using standard least squares linear regression are presented in Sect. 3 for the monthly, annual and seasonal (June−August) ERE. Section 4 presents estimates of ERE forcing by individual UVR drivers and their linear and non-linear superposition. Section 5 provides a discussion of the results.

## 2 Data

In this study, we used the following data collected at Belsk: ERE, TCO$_3$, sunshine duration (SunDur), daily global solar radiant exposure (G) and aerosol optical depth at 340nm (AOD$_{340\,nm}$). The data were archived in the IG Data Portal (Krzyścin, 2024). The instruments used and their operating periods are presented in Table 1.

**Table 1.** The Belsk instruments that were used in this study and their periods of operation.

| Data | Instrument | Operation period |
|---|---|---|
| Daily ERE | Robertson Berger Meter (290−350 nm) | 1976−1994 |
| | SL Biometer 501 A # 927 (280−400 nm) | 1993−1994 |
| | SL Biometer 501 A # 2011 (280−400 nm) | 1995−2013 |
| | Kipp-Zonen UV-AE-T # 30616 (280−400 nm) | 2014−present |
| $TCO_3$ | Dobson Spectrophotometer # 84 | 1963−present |
| SunDur | Campbell–Stokes sunshine recorder | 1966−1968, 1970−1973, 1975−now |
| G | Kipp CM 6, Sonntag PRM-2, Kipp&Zonen CM 11, and Kipp&Zonen CM 21 | 1966−now |
| $AOD_{340nm}$ | Sonntag pyrheliometers | 1976−2013 |
| | CIMEL CE 318-T | 2004−now |

## 2.1 Erythemal radiant exposures

Recently, daily ERE from various biometers operated at Belsk over the period 1976−2023 (as shown in Table 1) were homogenised by multiplying the measured (raw) data by the calibration coefficients, obtained as the smoothed ratio between the modelled erythemal irradiance at noon and the corresponding values from measurements on cloudless days. Tropospheric

Ultraviolet and Visible (TUV), available from http://acd.ucar.edu/UV/, was used to simulate erythemal irradiances at cloudless noon. TUV was developed by Madronich (1993) and version 5.4 was implemented here (TUV, 2025). Inputs to this model were time, $TCO_3$, $AOD_{340\,nm}$, Ångström coefficient α, surface albedo, single scattering albedo and aerosol asymmetry factor. The values of the last four variables were kept constant and equal to climatological values typical for rural sites. The raw, homogenised results from the biometers measurements and corresponding clear sky daily ERE (by TUV model) are available

on the IG Data Portal (Krzyścin, 2024). Here, in the case of days with missing UVR measurements, the gaps were filled with the corresponding monthly means of daily ERE (if the number of measurements in the month was not less than 14). If a whole month was missing (December 2014), the value was filled with the long-term monthly mean (2004−2023). The months of 1985 (June and July) were excluded from the analyses because of the lack of data, and replacing them with long-term monthly means seems to be incorrect, as UVR in these months contributes mostly to the annual ERE. The homogenisation procedure

of the Belsk UV measurements is described in detail by Krzyścin et al. (2025).

## 2.2 Ancillary data

Daily $TCO_3$ values are taken from the IG Data Portal. They are marked with quality flags. Flags 1−5 indicate measurements from Dobson spectrophotometer #84 (the most reliable values are marked with flag 1 and refer to direct sun measurements), flags 6−7 are from satellite measurements and flag 8 is from the Modern-Era Retrospective Analysis for Research and

100 Applications version 2 (MERRA-2) database (GMAO, 2025). Dobson results represent more than 90% of the time series. More details can be found in Krzyścin et al. (2015). SunDur has been measured with Campbell-Stokes sunshine recorders since 1966. As a predictor of cloud properties, SunDur is used as a relative value (percentage of daily duration, from sunrise to sunset). G has been measured since 1966 with various pyranometers (Table 1) calibrated to the Polish national standard (previously calibrated at the World Radiation Centre in Davos). In the years 1976−2013, $AOD_{340\,nm}$ was obtained from the

105 Linke turbidity factor measurements with Sonntag pyrheliometers (Posyniak et al., 2016), and since 2004 until now it has been measured within the Aerosol Robotic Network (AERONET) (AERONET, 2025) with CIMEL CE 318-T, data level 2.0. The clearness index (CI), i.e. the quotient of G and $G_0$ (clear sky value of daily global radiant exposure), was expressed in per cent, with $G_0$ from the fifth version of the global reanalysis of the European Centre for Medium-Range Weather Forecasts (ERA5,

2025) and (MERRA-2, 2025). For $G_0$, we compared measured G for cloudless sky conditions at Belsk with $G_0$ from MERRA-2 and ERA5. We found that the best fit was for the mean values of MERRA-2 and ERA5. Before 1980, we used ERA5, but the data was corrected with the mean bias.

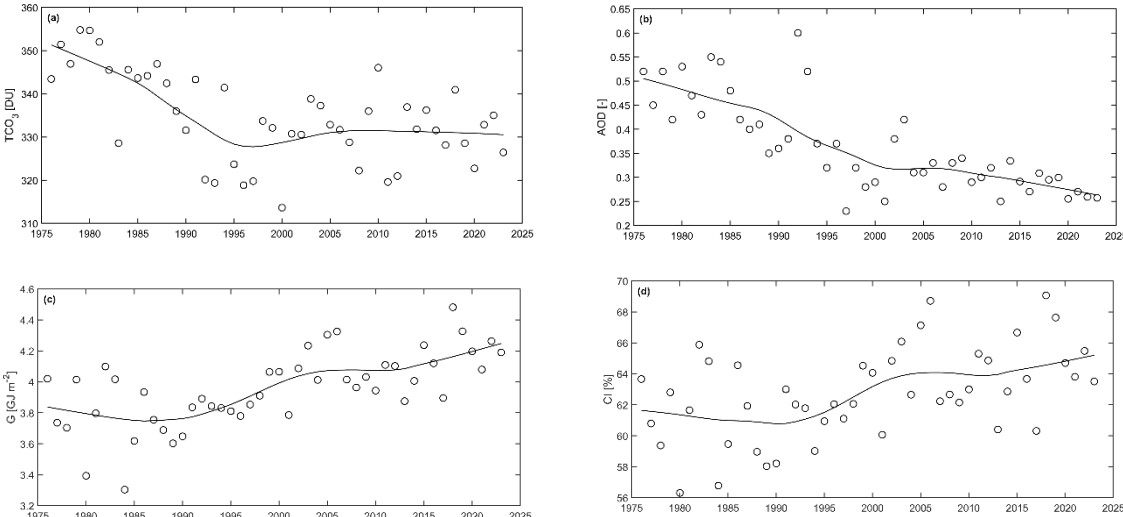

**Figure 1: The annual 1976−2023 time series of the basic UVR drivers: (a) TCO₃ in Dobson Unit, (b) AOD₃₄₀nm − dimensionless values, (c) yearly sum of G in GJ m⁻², (d) CI (G/G₀) – dimensionless value.**

The CI value, which is calculated on the basis of daily ERE, $CI_{ERE}$, is proportional to the CI value from global solar radiation. The conversion formula is often in the form: $CI_{ERE} = \alpha CI^{\beta}$, where the coefficients are derived empirically and may depend on solar zenith angle, resulting in a smaller $CI_{ERE}/CI$ ratio at lower solar elevation (e.g. Krzyścin et al. 2025). Figure 1 shows the annual means of TCO₃, AOD₃₄₀nm, CI, and the annual sum of G. These variables are fundamental drivers associated with clear sky (the first two variables) and cloud effects (the last two variables) on UVR. The time series of these drivers show considerable long-term variability. Their impact on monthly and annual ERE is presented later in the text (Sect. 4).

## 3 Trends in the UV measurements

This section presents the long-term variability of monthly, seasonal and annual ERE. Seasonal ERE refers to the summer months of June, July and August, which are the most important for human health, as the intensity of erythemal radiation tends to be high during these months and people spend more time outdoors. Our first guess was that a linear trend for the whole period 1976−2023 would not be applicable because of a trend shift in the annual ERE, visible in the smoothed data, that occurred between the years 1995−2015. The year 2000 was chosen as the trend-turning year that divided the whole observing period into two sub-periods (1976−1999 and 2000−2023) with equal length, and also to choose the same turning point for each time series. The effect of this selection on the trend values is discussed further in this Section.

The trends, expressed as a percentage of the 20-year average (2004−2023), are shown in Table 2 and Figs. 2−3. A linear trend is considered statistically significant if the doubled standard error (SE) of the slope coefficient is smaller than the absolute value of the slope. In such a case, the trend is shown in bold. The SE is multiplied by the autocorrelation factor $F = \sqrt{1 + R_{k+1})/(1 - R_{k+1})}$ if the autocorrelation coefficient for a time lag of one year, $R_{k+1}$, is greater than 0 (for $R_{k+1}<0$, $F=1$) (Weatherhead et al. 1998).

In the 1976−1999 sub-period, the positive linear trends were statistically significant in April (5.7% per decade), May (10% per decade), June (5.5% per decade) and July (8.6% per decade). However, in August (7.9% per decade) the trend was significant before adjusting for the 1-year lagged autocorrelation in the trend residuals. The upward seasonal (June−August) and annual

trends were statistically significant at 6.9% and 7.4% per decade, respectively. In the 2000−2023 sub-period, only the seasonal trend (3.1% per decade) was statistically significant.

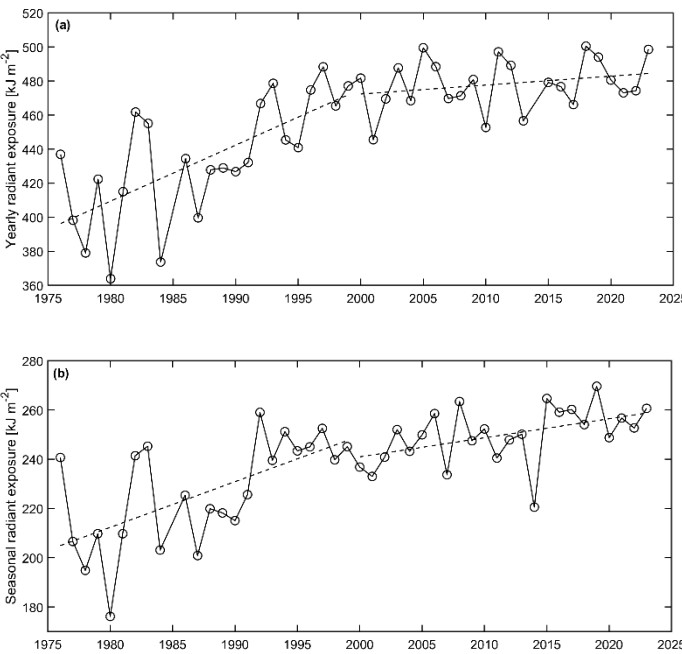

**Figure 2. Two disjoined linear regressions (for the 1976−1999 and 2000−2023 period) by the least-squares method applied to: (a) annual ERE, and (b) seasonal (June−July−August) ERE.**

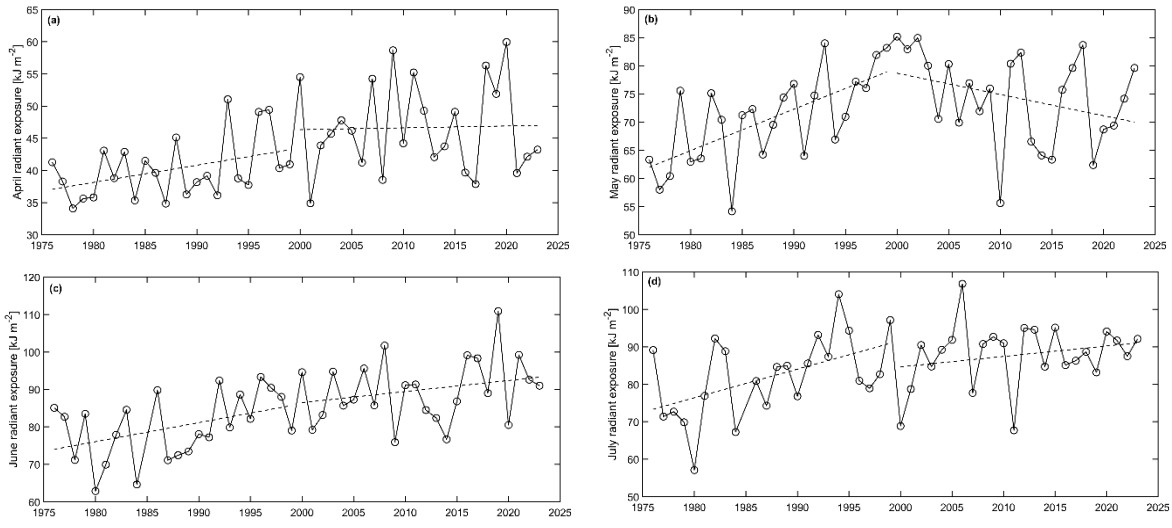

**Figure 3. The same as Fig.2 but for the monthly ERE: (a) - April, (b) - May, (c) - June, and (d)  July.**

In addition, ERE means from the last (2014−2023) and the first (1976−1985) decade of Belsk UVR observations were compared and the significance of the differences between them was checked using a two-sample $t$-test (Table 2). In most cases, the difference was statistically significant (March, April, June, July, August, September, December, seasonal and annual values) and in these cases it exceeded 10%. The largest differences were observed for June (~18%), April (~17%) and seasonal values (~16%). For annual values, the difference was ~14% and for other months where the difference was statistically significant the difference ranged from ~12% to ~15%. For January, February, May, October and November the difference was not significant and <10%.

**Table 2. Linear regression slopes (in % per year) in the first and second halves of the observations for monthly, seasonal (June−August) and annual erythemal radiant exposures with corresponding standard errors (SE$_{corr}$) adjusted for the one-year lagged autocorrelation (R$_{k+1}$) in the exposure time series. Relative difference (RelDiff) between the means for the last decade and the first decade, expressed as a percentage of the mean for the last decade. p-values are from $t$-tests for the difference between means. Statistically significant results are shown in bold.**

| Type of time series | Trends$_{1976-1999}$ [% per yr] | | Trends$_{2000-2023}$ [% per yr] | | RelDiff | p_Value |
|---|---|---|---|---|---|---|
| | Slope ($\pm SE_{corr}$) | $R_{k+1}$ | Slope ($\pm SE_{corr}$) | $R_{k+1}$ | [%] | |
| Jan | 0.53 ($\pm$0.45) | $-0.10$ | 0.00 ($\pm$0.41) | $-0.20$ | 0.57 | 0.92 |
| Feb | 0.04 ($\pm$0.44) | $-0.36$ | $-0.05$ ($\pm$0.36) | $-0.35$ | 8.24 | 0.19 |
| Mar | 0.74 ($\pm$0.43) | 0.13 | $-0.15$ ($\pm$0.46) | 0.26 | 12.04 | **0.03** |
| Apr | **0.57** ($\pm$0.27) | $-0.24$ | 0.06 ($\pm$0.45) | $-0.24$ | 16.58 | **0.01** |
| May | **1.02** ($\pm$0.25) | $-0.22$ | $-0.52$ ($\pm$0.33) | 0.04 | 9.16 | 0.06 |
| Jun | **0.55** ($\pm$0.26) | $-0.17$ | 0.33 ($\pm$0.28) | $-0.31$ | 17.97 | **<0.01** |
| Jul | **0.86** ($\pm$0.37) | 0.16 | 0.31 ($\pm$0.28) | $-0.16$ | 14.29 | **0.01** |
| Aug | 0.79 ($\pm$0.42) | 0.36 | 0.28 ($\pm$0.25) | $-0.20$ | 14.66 | **<0.01** |
| Sep | 0.43 ($\pm$0.38) | 0.05 | 0.24 ($\pm$0.45) | $-0.22$ | 14.30 | **0.02** |
| Oct | $-0.06$ ($\pm$0.34) | $-0.12$ | $-0.37$ ($\pm$0.47) | 0.10 | 2.50 | 0.70 |
| Nov | 0.63 ($\pm$0.42) | 0.07 | 0.07 ($\pm$0.36) | $-0.29$ | 2.95 | 0.60 |
| Dec | 0.68 ($\pm$0.35) | $-0.00$ | 0.35 ($\pm$0.66) | 0.12 | 13.13 | **0.02** |
| Yearly | **0.69** ($\pm$0.16) | $-0.07$ | 0.09 ($\pm$0.11) | 0.01 | 13.84 | **<0.01** |
| Seasonal | **0.74** ($\pm$0.24) | 0.14 | **0.31** ($\pm$0.12) | $-0.28$ | 15.91 | **<0.01** |

Independently, we analysed which trend-turning point would provide the best two-line regression fit to the observed ERE time series. We calculated a set of coefficients of determination, $R^2_T$, using two regression lines for the period 1976$-$T and T$-$2023, where T varied year by year from 1995 to 2015, and $T_{max}$ with the maximum $R^2$ ($R^2_{Tmax}$) was calculated. We found that although $R^2_{Tmax}$ is in some cases higher than $R^2_{2000}$, the difference in $R^2$ is not larger than 0.09, and for the series with statistically significant trends observed up to 2000, the difference does not exceed 0.05. The results are presented in Table 3.

**Table 3. Determination coefficient for two-slope regression with the turning point in 2000 ($R^2_{2000}$) and in $T_{max}$ ($R^2_{Tmax}$) with the best fit to the observed radiant exposure values. $T_{max}$ is selected from the 1995$-$2015 range.**

| Type of time series | $R^2_{2000}$ | $R^2_{Tmax}$ | $T_{max}$ |
|---|---|---|---|
| Jan | 0.05 | 0.05 | 1999 |
| Feb | 0.06 | 0.12 | 2013 |
| Mar | 0.14 | 0.19 | 2015 |
| Apr | 0.28 | 0.30 | 2009 |
| May | 0.31 | 0.36 | 2004 |
| Jun | 0.35 | 0.39 | 2009 |
| Jul | 0.25 | 0.28 | 1996 |
| Aug | 0.38 | 0.39 | 2002 |
| Sep | 0.21 | 0.30 | 1997 |
| Oct | 0.05 | 0.08 | 2015 |
| Nov | 0.06 | 0.10 | 2002 |
| Dec | 0.12 | 0.18 | 2006 |
| Yearly | 0.65 | 0.66 | 1996, 2007 |
| Seasonal | 0.56 | 0.56 | 1998, 2000 |

## 4 Factors affecting UV radiation

Several geophysical parameters characterising the attenuation of sunlight in the atmosphere have been monitored at Belsk in parallel with surface UV radiation measured with broadband biometers. These are: total column ozone, aerosol optical depth (at 340 nm), sunshine duration, and global solar irradiance. The last two parameters are used to derive relative sunshine duration and the clearness index as a percentage of daylight duration and hypothetical global solar irradiance under clear sky conditions, respectively. In addition, synthetic values of daily erythemal radiation exposure under clear sky conditions are calculated using the TUV model.

In this section, time series of monthly and annual ERE from the Belsk observations for the period 1976$-$2023 and the influence of different factors on UV radiation will be accessed, analysing the relative deviations of the observed and modelled ERE from

the reference level, i.e. the averaged values of monthly (Table 4) and annual EREs (477.5 and 684.5 kJ m$^{-2}$ for all sky and clear sky conditions, respectively) for the period 2004−2023. To reveal UV variations caused by a selected proxy and/or group of proxies, models are run with other proxies held at their reference values and only variations in the proxy of interest are allowed.

**Table 4. Mean 2004−2023 observed and modelled clear sky (by TUV model) monthly erythemal radiant exposure and corresponding mean values for potential proxies of UV variability including the daily sum of global solar irradiance (G), relative sunshine duration (rSun_Dur, daily sunshine duration in per cent of the day length), and clearness index (CI, daily sum of global solar irradiance in per cent of its synthetic clear sky representative).**

| Month | UV (kJ m$^{-2}$) | UV_clear (kJ m$^{-2}$) | TCO$_3$ (DU) | AOD − | G (MJ m$^{-2}$) | rSun_Dur (%) | CI (%) |
|-------|------|----------|------|-----|-----|----------|-----|
| Jan | 4.090 | 6.975 | 343.8 | 0.203 | 2.365 | 18.67 | 49.47 |
| Feb | 9.030 | 14.32 | 361.9 | 0.247 | 5.102 | 29.83 | 57.60 |
| Mar | 24.11 | 36.67 | 376.2 | 0.308 | 9.619 | 37.47 | 63.34 |
| Apr | 47.03 | 68.31 | 371.9 | 0.322 | 15.35 | 51.74 | 70.39 |
| May | 72.47 | 105.5 | 366.2 | 0.306 | 19.60 | 55.24 | 72.98 |
| Jun | 90.26 | 125.4 | 340.9 | 0.315 | 22.86 | 61.06 | 79.32 |
| Jul | 89.28 | 124.7 | 331.3 | 0.348 | 20.70 | 55.67 | 75.75 |
| Aug | 72.16 | 99.23 | 309.4 | 0.354 | 17.82 | 62.40 | 77.90 |
| Sep | 42.04 | 59.29 | 295.2 | 0.303 | 12.42 | 49.59 | 73.38 |
| Oct | 18.52 | 28.27 | 286.8 | 0.259 | 6.718 | 38.21 | 63.27 |
| Nov | 5.400 | 10.30 | 291.5 | 0.25 | 2.827 | 21.73 | 48.94 |
| Dec | 3.109 | 5.497 | 301.9 | 0.195 | 1.712 | 15.80 | 45.44 |

Table 5 shows the linear trends (1976−1999 and 2000−2023) in the time series of annual and seasonal means for the above-mentioned proxies for the effect of stratospheric ozone, aerosols and clouds on UVR. The negative trends in TCO$_3$ and AOD$_{340\,nm}$ were statistically significant in the first half of the observations for the annual and seasonal ERE, implying a positive trend in UVR. The negative aerosol trend continued in the second half of the observations but was about three times weaker than in the first sub-period. For the cloud effects, the positive trends are only found in the seasonal means for the second half of the observations. However, it will be discussed further in the text (Sect. 4.2) that changes in cloud parameters had a large impact on the UVR at Belsk.

**Table 5. The trends from the standard least squares regression of the UVR forcing factors and their standard errors for the periods 1976−1999 and 2000−2023. The statistically significant trends at the 2σ level are shown in bold.**

| Variable | Trend in % per decade ± 1 SE | | | |
|----------|:---------:|:--------:|:------:|:--------:|
| | 1976−1999 | | 2000−2023 | |
| | Annual | Seasonal | Annual | Seasonal |
| TCO$_3$ | **−3.7 ± 0.7** | **−3.6 ± 0.6** | 0.2 ± 0.7 | 0.2 ± 0.6 |
| AOD$_{340\,nm}$ | **−29.8 ± 9.8** | **−26.3 ± 8.9** | **−9.6 ± 3.7** | 2.5 ± 4.1 |
| G | 1.0 ± 1.4 | 1.7 ± 2.2 | 1.8 ± 1.2 | **3.4 ± 1.3** |
| CI | 0.3 ± 1.3 | −0.2 ± 2.2 | 0.7 ± 1.1 | **2.6 ± 1.2** |

**4.1 Clear-sky conditions**

A subset of clear sky days in the whole period of Belsk observations was extracted, assuming that the daily sum of total solar irradiance is close to its synthetic clear sky representative, taken from the ERA-5 and MERRA-2 reanalysis (Czerwińska et al., 2024, Krzyścin et al., 2025). It was arbitrarily set that the clearness index for such days should be greater than 95%. Fig. 4 shows a time series of the frequency of occurrence of clear sky days in the warm (April−September) and cold (October−March) sub-period of the year. Clear sky days are about twice as frequent in the former sub-period than in the latter. The pattern of long-term variability is similar in both cases, with a decrease in frequency at the beginning of the series that lasted until about 1990, and a further increase that is stronger in the period 1990−2005 than afterwards.

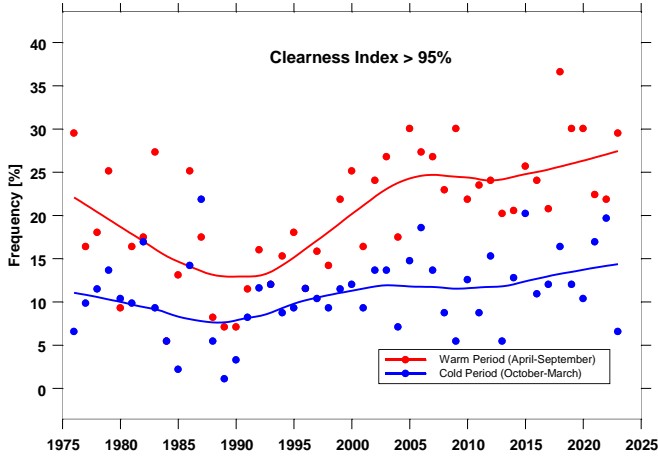

**Figure 4. Frequency of clear sky days in the warm (April−September) and cold (October−March) sub-period of the year for the period 1976−2023.**

TUV is a radiative transfer model that also provides biologically effective irradiances (here erythemal irradiance) throughout the cloudless day using the following set of input parameters: total column ozone, aerosol optical depth at a selected wavelength, and the Ångström coefficient. Other input parameters are usually set at a fixed level corresponding to typical values for the site. Taking into account climatology of the Belsk's aerosol characteristics (AERONET, 2025) values of 0.95 and 0.69 are taken for single scattering albedo and asymmetry factor, respectively. The Ångström coefficient was chosen to be equal to 0 on the assumption that the AOD values at 340 nm are representative of the whole UV-B range (300−315 nm) and the short wavelength part of the UV-A range (Jarosławski et al., 2003). Surface albedo is 0.03 representing a snowless surface covered by grass (Chadyšiene and Girgždys, 2008).

The long-term ozone effect on the clear-sky UV is derived from the version of the TUV model (denoted as $\text{TUV}_{O3}$) with input containing observed $TCO_3$ values, while all other UV forcing parameters are held constant, with AOD values equal to the monthly mean values calculated over the reference period (Table 4). Similarly, to quantify the aerosol effects, the $\text{TUV}_{AOD}$ model takes into account AOD at 340 nm from the measurements, but other input parameters are kept constant. Model $\text{TUV}_{O3+AOD}$ calculates the combined effects of ozone and aerosols, assuming that these effects are additive, i.e.,

$$\text{TUV}_{O3+AOD} = \text{TUV}_{O3} + \text{TUV}_{AOD} - \text{TUV}_{const} \qquad (1)$$

where $\text{TUV}_{const}$ is the TUV version using only constant values (the means for the reference period) for all input parameters. Model (1) is compared to the full TUV model (denoted as $\text{TUV}_{O3\&AOD}$) with the observed values of $TCO_3$ and AOD at 340 nm. The $\text{TUV}_{O3\&AOD}$ model may differ from $\text{TUV}_{O3+AOD}$ because the former model takes into account possible interactions between ozone and aerosols (e.g. cases with high $TCO_3$ and low AOD in air coming from the Arctic in early spring), which is neglected in the latter model, which by definition separates ozone effects from the aerosol effects.

Time series by all abovementioned versions of the TUV model are shown in Figure 5. From this Figure, it can be inferred that the long-term ozone and aerosols effects (as shown by the smoothed profiles obtained by LOcally WEighted Scatterplot Smoothing (LOWESS), Cleveland, 1979) are almost equal providing that the increase in the clear sky erythemal UV is of about 12% in the period 1976−1999. Afterwards, there is no trend in the $\text{TUV}_{O3}$ and $\text{TUV}_{AOD}$ values providing also trendless pattern of the $\text{TUV}_{O3+AOD}$ values.

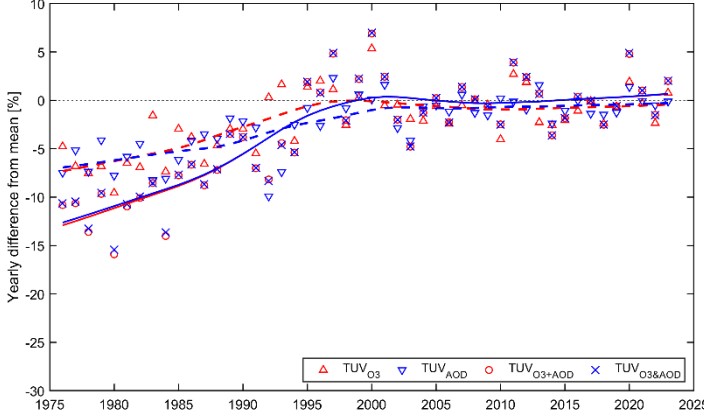

**Figure 5. Relative differences of the modelled annual erythemal radiant exposure from the pertaining clear-sky reference value of 684.5 kJ m$^{-2}$. The curves show the data smoothed by the LOWESS low-pass filter. The dashed curve and triangles in red refer to the TUV$_{O3}$ model, and the dashed curve and triangles in blue refer to the TUV$_{AOD}$ model. The solid curve and circles in red refer to the TUV$_{O3+AOD}$ model, and the solid curve and crosses in blue refer to the TUV$_{O3\&AOD}$ model.**

It is worth noting that the long-term pattern of annual EREs by the TUV$_{O3+AOD}$ and TUV$_{O3\&AOD}$ models are close to each other, which means that the effects due to the interaction of these basic clear-sky UV proxies could be omitted from trend analyses. This also makes it possible to construct a simple UV regression model that relates the relative changes in the monthly erythemal exposure from the TUV$_{O3\&AOD}$ model, $\Delta TUV_{O3\&AOD}(i, j)$, to the linear combination of the relative changes in the measured TCO$_3$ and AOD$_{340}$ nm values:

$$\Delta TUV_{O3\&AOD}(i, j) = \alpha_0(j) + \beta_0(j) \times \Delta TCO_3(i, j) + \gamma_0(j) \times \Delta AOD_{340\,nm}(i, j) \qquad (2)$$

where $j$ denotes calendar month and $i$ is the year ($i$=1976 for the first year of observations and 2023 for the last year). Estimates of the model coefficients with their standard errors are presented in Table 6, together with the standard deviation of the regression model residuals and the coefficients of determination (the squared value of the correlation coefficient between model (2) and TUV$_{O3\&AOD}$).

**Table 6. Regression coefficients of the model (2) and their standard errors (SE) for calendar months. Corresponding standard deviation (SD) of the differences and squared correlation coefficient ($R^2$) between model (2) and TUV$_{O3\&AOD}$ (TUV model using measured TCO$_3$ and AOD$_{340\,nm}$) values. Value in bold indicates statistically significant value at 2SE level. Dimensionless values are denoted as (−).**

| Month | $\alpha_0 \pm$ SE (%) | $\beta_0 \pm$ SE (−) | $\gamma_0 \pm$ SE (−) | SD (%) | $R^2$ (−) |
|---|---|---|---|---|---|
| Jan | **1.112** ± 0.372 | **−0.831** ± 0.033 | **−0.084** ± 0.004 | 1.580 | 0.960 |
| Feb | **2.749** ± 0.369 | **−0.966** ± 0.036 | **−0.109** ± 0.006 | 1.740 | 0.964 |
| Mar | 0.389 ± 0.231 | **−1.032** ± 0.036 | **−0.122** ± 0.006 | 1.335 | 0.970 |
| Apr | −0.116 ± 0.183 | **−1.082** ± 0.033 | **−0.115** ± 0.006 | 1.107 | 0.981 |
| May | **−0.940** ± 0.105 | **−1.103** ± 0.022 | **−0.098** ± 0.003 | 0.594 | 0.992 |
| Jun | **0.311** ± 0.072 | **−1.076** ± 0.020 | **−0.096** ± 0.003 | 0.416 | 0.996 |
| Jul | **−0.806** ± 0.071 | **−1.045** ± 0.021 | **−0.108** ± 0.003 | 0.457 | 0.993 |
| Aug | **−0.299** ± 0.109 | **−1.062** ± 0.028 | **−0.121** ± 0.004 | 0.654 | 0.988 |
| Sep | 0.121 ± 0.130 | **−1.041** ± 0.030 | **−0.110** ± 0.003 | 0.737 | 0.985 |
| Oct | **0.844** ± 0.279 | **−0.952** ± 0.056 | **−0.094** ± 0.005 | 1.367 | 0.949 |
| Nov | **0.678** ± 0.302 | **−0.874** ± 0.056 | **−0.099** ± 0.005 | 1.433 | 0.937 |
| Dec | **1.917** ± 0.280 | **−0.842** ± 0.027 | **−0.071** ± 0.003 | 1.143 | 0.981 |

The results presented in Table 6 confirm that the linear regression of monthly ERE on TCO$_3$ and AOD$_{340\,nm}$ is in perfect agreement with the TUV$_{O3\&AOD}$ model, as all monthly coefficients of determination are above 95% and for the spring and summer months this coefficient is even above 98%. It can also be seen that under cloudless sky conditions in the warm (snow-free) season, a 1% decrease (increase) in TCO$_3$ results in a ~1% increase (decrease) in erythemal radiation exposure.

Correspondingly, this rate is about 10 times smaller for changes in $AOD_{340\,nm}$. Slightly lower, but still statistically significant, rates occur during the cold season. These estimates are consistent with earlier findings by Krzyścin and Puchalski (1998).

### 4.2 All-sky conditions

#### 4.2.1 Non-linear model

The machine learning (ML) approach is implemented to model daily EREs as a function of all available proxies of the surface UVR which have been monitored at Belsk since 1 January 1976 including $TCO_3$, $AOD_{340\,nm}$, relative daily sunshine duration (SunDur) in per cent of the day length, the daily sum of global solar irradiance (G), clearness index (CI i.e., relative G in per cent of its clear sky representative). These proxies together with measured ERE are archived in the IGF PAS data portal (Krzyścin, 2024).

The ML models were trained using the Regression Learner App in Matlab version 2018a (https://www.mathworks.com/help/stats/regressionlearner-app.html). This tool allows multiple regression models to be trained, their validation errors compared and the best fit selected. The regression models available in this version are: linear regression models (linear, interaction linear, robust linear, stepwise linear), regression trees (fine tree, medium tree, coarse tree), support vector machines (linear, quadratic, cubic, fine Gaussian, medium Gaussian, coarse Gaussian), ensemble of trees (boosted and bagged) and Gaussian Process Regression (GPR) models (rational quadratic, squared exponential, maternal 5/2 and exponential). GPR models are non-parametric, probabilistic and kernel-based. These types of models use a probability over a space of functions as an answer. In the Regression Learner App, the flexibility of the presets is automatically chosen to give the smallest error. Using this approach, we select two GPR models from all available models for further calculations. The best fit for all predictors was a GPR model with an exponential kernel function (which determines the correlation in the response), while in the model with a limited number of predictors ($TCO_3$, G and relative SunDur) the best fit was GPR with a maternal 5/2 kernel function. Other parameters (kernel mode, sigma mode and base function) were automatically set to optimise the model.

Further in the text, the model built from all available daily values of the proxies will be denoted as $GPR_{all}$. In addition, the various output of the $GPR_{all}$ model will be examined using:

- $GPR_{const}$ with all proxies fixed to their daily means for the period $2004-2023$
- $GPR_{clear}$ with daily values of $TCO_3$, $AOD_{340\,nm}$. Other proxies are fixed to the daily means for the period $2004-2023$
- $GPR_{cloud}$ with daily values of relative SunDur, G, CI. Other proxies are fixed to the daily means for the period $2004-2023$
- $GPR_{clear+cloud} = GPR_{clear} + GPR_{cloud} - GPR_{const}$

Model $GPR_{clear}$ is for searching a contribution of combined effects of clear sky proxies on the measured daily ERE whereas $GPR_{cloud}$ is for combined effects due to the cloud proxies. The comparison of $GPR_{clear+cloud}$ and $GPR_{all}$ allows us to see if the effects of ozone and aerosols can be separated from the effects of clouds (i.e. the interaction between these groups of variables can be ignored) when discussing the sources of monthly and annual ERE variability.

A version of the GPR model using a limited number of proxies, $GPR_{TCO3* \& SunDur \& G,}$ is also constructed, using $TCO_3$ values from Multi Sensor Reanalysis version 2 (van der A, 2015), hereafter referred to as $TCO_{3*}$, and only two cloud characteristics of clouds (SunDur and G) from the measurements at Belsk. $GPR_{TCO3* \& SunDur \& G}$ is to reveal a loss of the ML model accuracy when the limited number of proxies is only available from daily monitoring. For a weather station equipped with a Campbell-Stokes recorder and pyranometer, this UV proxy set may be available.

The performance of the most ($GPR_{all}$) and least ($GPR_{TCO3* \& SunDur \& G}$) extended models is shown in Table 7. The typical descriptive statistics are taken into account: mean relative error (MRE), mean absolute error (MAE), root mean square error (RMSE), and standard deviation (SD) of the relative differences (in per cent of the observed values) between modelled and

observed values, and R denotes the Pearson's correlation coefficient between these values. Formal definitions of the statistics can be found in our recent paper (Krzyścin et al., 2025).

**Table 7. Descriptive statistics of the most ($GPR_{ALL}$) and least ($GPR_{TCO3* \& SunDur \&G}$) extended GPR model calculated using the data collected over the period 1976−2023.**

| Month | $GPR_{ALL}$ | | | | | $GPR_{TCO3* \& SunDur \&G}$ | | | | |
|---|---|---|---|---|---|---|---|---|---|---|
| | MRE [%] | MAE [%] | RMSE [%] | SD [%] | R | MRE [%] | MAE [%] | RMSE [%] | SD [%] | R |
| Jan | −0.05 | 2.24 | 2.88 | 2.91 | 0.98 | −0.33 | 4.21 | 5.50 | 5.55 | 0.93 |
| Feb | −0.49 | 2.36 | 3.46 | 3.47 | 0.97 | −0.62 | 3.80 | 5.43 | 5.45 | 0.93 |
| Mar | −0.62 | 2.26 | 5.10 | 5.11 | 0.96 | −0.64 | 3.17 | 6.29 | 6.32 | 0.94 |
| Apr | 0.14 | 0.98 | 1.40 | 1.40 | >0.99 | 0.21 | 1.28 | 1.69 | 1.69 | 0.99 |
| May | −0.04 | 1.17 | 1.56 | 1.58 | >0.99 | −0.07 | 1.81 | 2.40 | 2.42 | 0.98 |
| Jun | −0.04 | 1.22 | 1.64 | 1.65 | >0.99 | −0.07 | 1.91 | 2.60 | 2.63 | 0.98 |
| Jul | 0.02 | 1.31 | 1.73 | 1.75 | >0.99 | 0.02 | 2.07 | 2.71 | 2.74 | 0.98 |
| Aug | −0.26 | 1.66 | 2.96 | 2.98 | 0.98 | −0.44 | 2.75 | 4.18 | 4.20 | 0.95 |
| Sep | −0.13 | 1.22 | 1.56 | 1.57 | >0.99 | −0.18 | 2.12 | 2.70 | 2.72 | 0.99 |
| Oct | 0.19 | 1.45 | 1.82 | 1.83 | >0.99 | 0.19 | 2.48 | 2.95 | 2.97 | 0.98 |
| Nov | −0.70 | 2.11 | 3.53 | 3.50 | 0.97 | −0.70 | 3.60 | 5.91 | 5.94 | 0.92 |
| Dec | 0.42 | 3.31 | 4.80 | 4.84 | 0.95 | 0.42 | 6.25 | 8.50 | 8.59 | 0.79 |

The performance of the two models is very similar, as shown by the small differences in the descriptive statistics, especially in the warm subperiod of the year, where the SD and MAE are only about 1−1.5 per cent point (%p) higher for the version of the model with the fewest proxies. For both models, the model-observation correlation coefficients are very high, exceeding 0.94 for all monthly data within the warm period. In April, the performance of the two models is only slightly different. The difference is less than 0.3 %p. Larger differences are found in the cold season, especially in December, when the SD and MAE of the least extended GPR model are about 5 and 3 %p higher than for the $GPR_{ALL}$ model using all available proxies.

For all GPR models, the monthly sums of the daily ERE are calculated and expressed as relative differences from the monthly 2004−2023 references of the ERE (Table 4). The annual EREs are calculated by adding the monthly sums, which are finally expressed as relative differences from the annual reference of 477.5 kJ m$^{-2}$. Fig. 6 shows the 1976−2023 time series of the relative differences in per cent of the annual ERE reference for the observed and set of modelled GPR ($GPR_{clear}$, $GPR_{cloud}$, $GPR_{clear+cloud}$, and $GPR_{all}$).

Analysing the long-term variability in these time series, it can be seen that the annual ERE increases by about 15% in the period 1976−2023, i.e. close to the rate found in the synthetic clear-sky data (Fig. 5). This increase is due to the linear superposition of the ozone and aerosols ($GPR_{clear}$) and cloud ($GPR_{cloud}$) effects, as the smoothed $GPR_{clear+cloud}$ time series differs only slightly from the smoothed annual ERE time series based on measurements, which is in perfect agreement with that from the full GPR model ($GPR_{all}$). The forcing effects of combined ozone and aerosols impact with clouds cancelled each other out at the beginning of the observations, but from about 1986, they started to act in a unidirectional way, causing a fast increase that stopped around 2000. The combined ozone and aerosols effect levelled off a few years before 2000, but the cloud effect levelled off a few years after 2000. Subsequently (around 2010−2015), cloud forcing caused the upward trend to reappear around 2010−2015, but the rate of increase is rather small, comparable to that seen in the 1980s and 1990s. This pattern of long-term changes is generally consistent with that found in the smoothed time series of the observed and $GPR_{all}$ data. However, a complete compensation of the ozone & aerosols and cloud effects found in $GPR_{clear+cloud}$, which occurred until about 1986, is not supported by $GPR_{all}$ and observed data, as they rather show a slight increase in the annual ERE during this period.

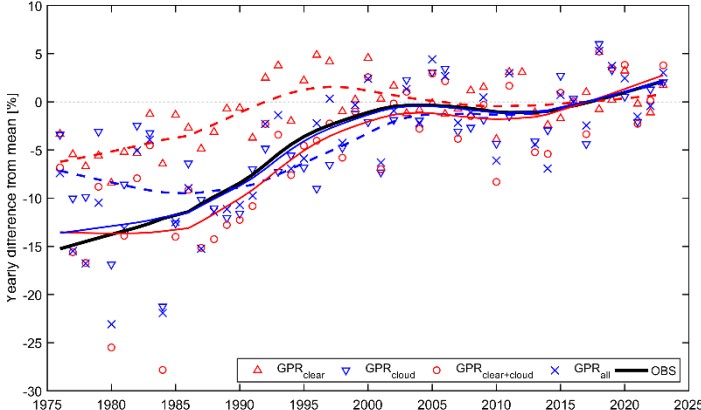

**Figure 6. Relative differences of annual erythemal radiant exposures from the 2004−2023 reference for the observed (black solid curve) data and modelled by various versions of the GPR model: *GPR*ₑₗₑₐᵣ (dashed curve and triangle up in red), *GPR*ₑₗₒᵤ𝒹 (dashed curve and triangle down in blue), *GPR*ₑₗₑₐᵣ₊ₑₗₒᵤ𝒹 (solid curve and open circles in red), and *GPR*ₐₗₗ (solid curve and crosses in blue). The curves show the data smoothed by a LOWESS low-pass filter.**

Figure 7 shows the long-term variability of the annual ERE by the four time series taken from $GPR_{TCO3* \& SunDur \&G}$ output (GPR$_{TCO3*}$, GPR$_{SunDur*G}$, GPR$_{TCO3*+SunDur*G}$, and $GPR_{TCO3* \& SunDur \&G}$) that corresponds to the four time series by the most advanced *GPR*ₐₗₗ model (*GPR*ₑₗₑₐᵣ, GPR$_{cloud}$, *GPR*ₑₗₑₐᵣ₊ₑₗₒᵤ𝒹, and *GPR*ₐₗₗ). The $GPR_{TCO3* \& SunDur \&G}$ model is for annual ERE estimates for a station with low-cost equipment, i.e. a pyranometer and a Campbell-Stokes instrument, as TCO$_3$ comes from satellite observations, which is indicated by an asterisk after TCO$_3$. It can be seen that the long term patterns shown in Fig.6 are almost reproduced, but the $GPR_{TCO3* \& SunDur \&G}$ model gave only slightly larger (lower) differences between the observations up to about 1985 (from 1985 up to 2005−2010).

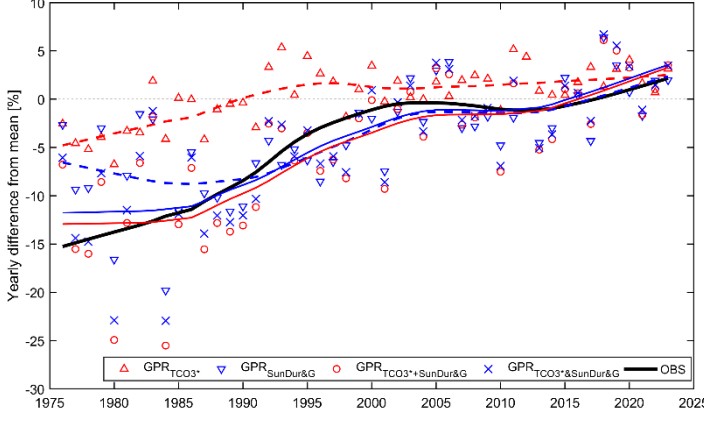

**Figure 7. Relative differences of annual erythemal radiant exposures from the 2004−2023 reference for the observed (black solid curve) data and modelled by four versions of the *GPR*$_{TCO3* \& SunDur \&G}$ output: *GPR*$_{TO3*}$ (dashed curve and triangle up in red), *GPR*$_{SunDur\&G}$ (dashed curve and triangle down in blue), *GPR*$_{TCO3*+SunDur\&G}$ (solid curve and open circles in red), and *GPR*$_{TCO3*\&SunDur\&G}$ (solid curve and crosses in blue). The curves show the data smoothed by a LOWESS low-pass filter.**

### 4.2.2 Linear regression model

To calculate the direct UV response to the basic forcing factors (TCO$_3$, aerosols, and clouds), a standard linear regression (*SLR)* of the observed monthly ERE on only two proxies is examined further in this section, i.e. TUV$_{O3\&AOD}$ (clear-sky representative of UV radiation by the TUV model using observed values of TCO$_3$ and AOD$_{340\ nm}$) and CI. The former proxy is intended to account for clear sky effects and the latter for the cloud effects on all-sky monthly ERE over the whole period (1976−2023) of the UVR observations at Belsk. Further, this model is denoted as *SLR*ₑₗₑₐᵣ&ₑₗₒᵤ𝒹 and uses the fractional deviations of TUV$_{O3\&AOD}$ and CI from the references (shown in Table 1) as the explaining variables:

$$\Delta SLR_{clear\&cloud} (i, j) = \alpha_1(j) + \beta_1(j) \times \Delta TUV_{O3\&AOD} (i, j) + \gamma_1(j) \times \Delta CI(i, j) \tag{3}$$

**Table 8. Regression coefficients of the model (3) and their standard errors (SE) for calendar months. Corresponding standard deviation (SD) of the differences and squared correlation coefficient ($R^2$) between model (3) and the monthly ERE from the measurements. Bolded values indicate a statistically significant regression coefficient at the 2SE level. Dimensionless values are denoted as (−).**

| Month | $\alpha_1 \pm SE$ (%) | $\beta_1 \pm SE$ (−) | $\gamma_1 \pm SE$ (−) | $\gamma_1 \beta_1^{-1}$ (−) | SD (%) | $R^2$ (−) |
|-------|-----------------------|----------------------|-----------------------|-----------------------------|--------|-----------|
| Jan | −1.684 ± 1.507 | **0.354** ± 0.168 | **0.729** ± 0.085 | 2.06 | 8.851 | 0.645 |
| Feb | 2.227 ± 1.343 | **0.635** ± 0.136 | **0.754** ± 0.095 | 1.19 | 8.324 | 0.641 |
| Mar | 0.844 ± 0.973 | **0.701** ± 0.122 | **0.763** ± 0.073 | 1.09 | 6.213 | 0.780 |
| Apr | 0.357 ± 0.604 | **0.782** ± 0.073 | **0.820** ± 0.047 | 1.05 | 3.591 | 0.940 |
| May | **2.348** ± 0.566 | **0.827** ± 0.084 | **0.871** ± 0.054 | 1.05 | 3.589 | 0.903 |
| Jun | **2.964** ± 0.727 | **0.669** ± 0.094 | **0.842** ± 0.058 | 1.26 | 3.772 | 0.886 |
| Jul | −0.193 ± 0.610 | **0.634** ± 0.103 | **0.778** ± 0.048 | 1.23 | 3.675 | 0.896 |
| Aug | 0.868 ± 0.771 | **0.878** ± 0.116 | **0.847** ± 0.079 | 0.96 | 4.496 | 0.844 |
| Sep | 0.716 ± 0.710 | **0.723** ± 0.121 | **0.813** ± 0.049 | 1.12 | 4.109 | 0.928 |
| Oct | 0.273 ± 0.836 | **0.359** ± 0.140 | **0.720** ± 0.055 | 2.01 | 5.207 | 0.845 |
| Nov | **4.060** ± 1.279 | **0.465** ± 0.213 | **0.653** ± 0.085 | 1.40 | 7.772 | 0.647 |
| Dec | −1.373 ± 2.283 | **0.933** ± 0.234 | **0.433** ± 0.106 | 0.46 | 12.530 | 0.486 |

The regression constants by the least-squares approach are shown in Table 8. The determination coefficients ($R^2$) are high (approximately 0.8 to 0.9) in the March−October sub-period of the year. Thus, the annual ERE can be also correctly estimated as these months contribute to ~95% of the annual ERE. The highest and lowest $R^2$ values of 0.94 and 0.49 are in April and December, respectively. In addition, it is worth mentioning that in the warm period of the year, the rates of change in the monthly ERE due to 1% change in the explaining variables are almost equal (i.e. $\gamma_1 \beta_1^{-1}$ ~1) and in the ranges (0.63−0.88) and (0.78−0.87) for the clear and cloud effects, respectively.

Having values of the regression coefficients, it will be easy to separate the clear sky effects from the cloud effects using the truncated versions of the model (3), which follow the previous concept (applied to the GPR$_{all}$ model) of allowing variations in one variable but keeping fixed values for other variables:

$$\Delta SLR_{const} (i, j) = \alpha_1(j) \tag{4}$$

$$\Delta SLR_{clear} (i, j) = \alpha_1(j) + \beta_1(j) \times \Delta TUV_{O3\&AOD} (i, j) \tag{5}$$

$$\Delta SLR_{cloud} (i, j) = \alpha_1(j) + \gamma_1(j) \times \Delta CI(i, j) \tag{6}$$

$$\Delta SLR_{clear+cloud} = \Delta SLR_{clear} + \Delta SLR_{cloud} - \Delta SLR_{const} \tag{7}$$

Here, from the definition of the linear model, $SLR_{clear\ \&\ cloud} = SLR_{clear+Cloud}$.

The pattern of long-term UV forcing variability shown in Fig. 8 is in general agreement with that obtained by the most and least advanced GPR models, as shown in Figs. 6 and 7, respectively. The cloud effects are underestimated in the model (3), ranging from −2% to −5% over the period 1976−1990, while the corresponding change in the GPR$_{all}$ model is −6% to −10%. There is less agreement between the results of model (3) and the observations, as indicated by the larger SD values in Table 8 compared to the corresponding values in Table 7. The long-term pattern of annual ERE from the model (3) starts to correspond

exactly to measurements from about 2000−2005 onwards (similarly to that from the less advanced GPR model, Fig.7), but this happened earlier (~1985) in GPR$_{all}$ model (Fig.6).

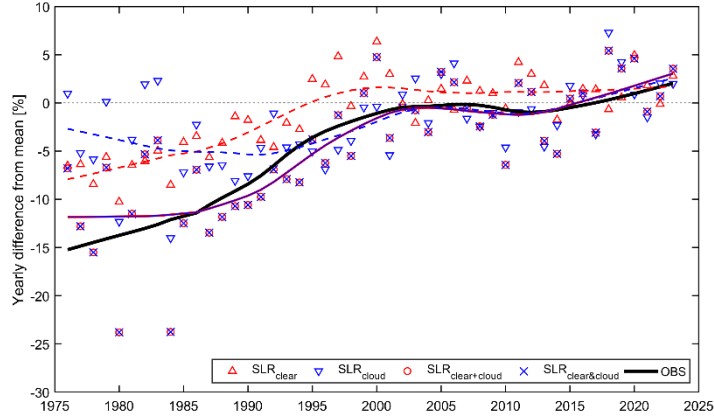

**Figure 8. Similar to Fig.6 but time series show various versions of the standard linear regression (SLR) using the clear-sky ERE and clearness index as the explaining variables.**

### 4.2.3 Ranking of the drivers in the annual exposure models

The ERE drivers in the model discussed in Sect.4 are divided into two categories, i.e. TCO$_3$ and AOD$_{340\,nm}$ for the clear sky

cases simulated by the TUV model, and cloud (SunDur, G, and CI) drivers for all sky cases considered in the advanced, simplified GPR, and SLR model. To compare the effects of selected drivers of ERE, the strength coefficient of driver X in a model MOD (SC_MOD$_X$) is proposed as the square root of the mean square of the differences between MOD$_x$ and MOD$_{const}$ values. Namely, the X factor can be "clear" and "cloud" in the GPR model for the all-sky ERE (Sect.4.2.1) but "TCO$_3$" and "AOD" in the TUV model for the clear-sky ERE (Sect.4.1). Table 9 shows SC_MOD$_X$ values for all models and individual

drivers used in Sect. 4.

**Table 9. Strength coefficients (SC_MOD$_X$ ) of the X drivers in the MOD$_x$ model of the annual ERE and their ratios between the second and first driver for the following pairs: X= {AOD, TCO$_3$}, {Clear, Cloud}, {TCO$_3$, SunDur&G}, and {Clear, Cloud} for the TUV, advanced GPR, simplified GPR, and SLR model, respectively. The SC_MOD$_X$ is in per cent of the annual reference value (i.e. 684.5 kJ m$^{-2}$ and 477.5 kJ m$^{-2}$, for the TUV and all-sky models, respectively). The results are for the 1976−2023, 1976−1999, and**

**2000−2023 period.**

| | | | | MOD$_x$ | | | | | | | |
|---|---|---|---|---|---|---|---|---|---|---|---|
| TUV$_{AOD}$ | TUV$_{TCO3}$ | ratio | GPR$_{Clear}$ | GPR$_{Cloud}$ | ratio | GPR$_{TCO3}$ | GPR$_{SunDur\&G}$ | ratio | SLR$_{clear}$ | SLR$_{Cloud}$ | ratio |
| | | | | | 1976−2023 | | | | | | |
| 3.19 | 3.24 | 1.01 | 4.52 | 8.75 | 1.93 | 3.09 | 7.81 | 2.53 | 4.52 | 5.48 | 1.21 |
| | | | | | 1976−1999 | | | | | | |
| 4.26 | 4.03 | 0.95 | 5.72 | 11.47 | 2.00 | 4.08 | 10.21 | 2.50 | 6.07 | 7.01 | 1.16 |
| | | | | | 2000−2023 | | | | | | |
| 1.52 | 2.17 | 1.43 | 2.86 | 4.38 | 1.53 | 1.59 | 3.97 | 2.50 | 2.01 | 3.30 | 1.64 |

The effect of the aerosol driver is as strong as that of the ozone driver for the sub-period 1976−1999 and for the whole period 1976−2023. The ozone driver dominates the aerosol driver in the last sub-period 2000−2023 (see the ratio of 1.43 in Table 9). The effects of the cloudy drivers in both GPR models are stronger than the clear sky drivers for the sub-periods 1976−1999,

2000−2023 and for the whole period 1976−2023 (the ratios are in the range [1.5, 2.5]). The same patterns are observed for the SLR drivers in the sub-periods 1976−1999, 2000−2023 and for the whole period 1976−2023, but the dominance of the cloud driver is less evident with the strength coefficients in the range [1.2, 1.6].

**5 Discussion**

In Europe, UVR measurements mainly come from broadband instruments, which are grouped in national UV monitoring
networks to inform the public about the danger of strong UVR through a UV index (Schmalwieser et al., 2017). This is also
the case in Poland, where there are several stations with regular UVR monitoring. Among them, Belsk has the longest time
series (48 years) (Krzyścin et al., 2025). The previous trend analysis of the Belsk 1976−2008 UVR data showed a statistically
significant increase in annual ERE of 5.6 ± 0.9% per decade for this period (Krzyścin et al., 2011). Using reconstructed annual
ERE based on statistical models, den Outer et al. (2010) calculated trends of 5.2 ± 1.3% and 5.8 ± 1.5% per decade at Hradec
Kralove (50.2° N, 15.8° E), Czech Republic and Lindenberg (52.2° N, 14.1° E), Germany, respectively, for the period
1980−2006. The trend analyses at these neighbouring stations of Belsk showed a slightly less positive trend than the present
estimate of 6.9 ± 1.6% per decade for the period 1976−1999. Such a difference seems to be a local effect, not caused by the
different periods used for trend estimations, as the trend at Belsk calculated exactly for the period 1980−2006 was 7.0 ± 1.2%
per decade, i.e. almost the same as for the period 1976−1999. Fountoulakis et al. (2020) found that local effects mainly related
to the changes in aerosol load, albedo and cloud cover changes in Europe, were important in shaping the UVR trend. In the
case of Reading (51.4° N, 0.9° W), UK, and Sodankylä (67.4° N, 26.6° E), Finland, downward trends in the 307.5nm irradiance
were even found between 1996 and 2017. In Central Spain, Bilbao et al. (2011), based on a reconstructed series of the erythemal
irradiance, recorded upward trends of 3.5% and 4.1% per decade for summer and autumn in the period 1991−2010.

For the shorter period in the 1980s and 1990s, the annual ERE trend was even larger. From the reconstructed daily ERE,
Čížková et al. (2018) calculated trends in successive decades (1964−1973, 1974−1983, 1984−1993, 1994−2003, 2004−2013)
and only the 1984−1993 trend appeared statistically significant at 14.9 ± 6.1 % per decade. The corresponding trend at Belsk
was found to be also slightly higher at 19.4 ± 4.2% per decade. Such a strong trend between 1984 and 1993 was due to the
overlapping clear sky and cloud effects, both of which caused UVR to increase (Figs. 6−7).

UVR measurements at stations in Eastern Europe have not shown statistically significant annual ERE trends in the current
century, including at Moscow (55.8° N, 37.9° E), Russia, in the period 1999−2015 (Chubarova et al., 2018), and at Tõravere
(58.37° N, 26.5° E), Estonia, in the period 2004−2016 (Aun et al., 2019). This disappearance of the trend was due to the lack
of trends in the main UV forcing factors, i.e. $TCO_3$, AOD and G. The same trendless pattern of annual ERE was found at Belsk
without trends in these forcing factors since 2000 (Table 5). Zerefos et al. (2012), who analysed the long-term variability of
UV solar irradiances at 305 nm and 325 nm over selected sites in Canada, Europe and Japan, pointed out that for all of these
sites, the trendless period began around 2007. However, the continuation of the upward trend (3.1 ± 1.2% per decade) since
2000 was found in Belsk only in the seasonal (June−August) ERE due to lower attenuation of downward solar radiation, as
the positive trends in G and CI were statistically significant at 3.4 ± 1.3% and 2.6 ± 1.2% per decade for the period 2000−2023.
In addition, the frequency of days with clear skies (i.e. higher G) increased during this period (Fig. 4).

The idea behind the proposed method to identify the long-term variability sources is to construct an individual time series of
ERE due to each selected proxy (or set of proxies), holding other proxies at their reference levels (means for the last 20 years).
Using this method, it was found that superimposed clear sky and cloud proxy effects between 1985 and 1999 were the source
of the positive trend in annual ERE in the first half of the observations (Fig. 6). Prior to 1985, clear sky and cloud effects
almost cancelled each other out. The clear sky and cloud effects stabilised after about 1996 and 2005 respectively, but the
latter proxy seems to force a positive trend in the annual ERE again since about 2015 due to increasing cloud transparency.
This conclusion aligns with other studies. Fountoulakis et al. (2021) found that in 2006-2020, the levels of UV irradiance at
Aosta, Rome and Lampedusa (Italy) increased mainly due to changes in cloud clouds or/and aerosol. Lorenz et al. (2024) also
attribute the increasing trend in erythemal UV radiation in Dortmund (Germany) in 1997-2022 to a change in cloud cover as
the primary driver. In the reconstructed series in Novi Sad (Serbia) in 1971-2018, Malinović-Milićević et al. (2022) found that
there were no significant trends in the last two decades, but there was an increase in the number of days with high daily radiant
exposures. The authors also stated that the cloud cover had more influence than $TCO_3$.

The uncertainty of the homogenisation procedure of the Belsk's UV measurements can be assessed by comparing the descriptive statistics of the differences between the regression model and the observed daily EREs from the KZ meter (operated from 2014 to the present day with no required adjustments, as evidenced by the excellent agreement with co-located Brewer spectrophotometer observations; Krzyścin et al., 2025) with the corresponding statistics of the model-observation differences using the homogenised RB and SL501A measurements (1976–1992 and 1993–2013, respectively). The model was trained using data from 2014 to 2023 and applied to pre-2014 periods. For full-year data, the mean relative difference between the model and the KZ observations is 1.4%, whereas it is between −2.7% and 1.6% for the RB and SL501A meters, respectively. The respective standard deviations were found to be 10.4%, 13%, and 14.6% (see Table 4 in Krzyścin et al., 2025). These descriptive statistics are only a few percentage points higher than those obtained for the KZ measurements. Thus, the homogenisation procedure for the period 1976–2013 resulted in a slight increase in uncertainty due to differences in the individual instrumental characteristics of the RB and SL501A meters, as well as the limited aerosol input data (annual mean AOD) used in the regression model, which was derived from sparse Sonntag pyrheliometer measurements in that period.

To account for the effects of aerosols, the daily-averaged AOD at 340 nm was used with an Ångström exponent (AE) of 0 across the UV range, while the other features were held constant. This approach is consistent with the homogenisation procedure for Belsk's UV measurements from 1976 to 2023, as proposed by Krzyścin et al. (2025). Our previous study examined Brewer and the Cimel Sun photometer measurements at Belsk and showed that AE was close to zero in modelling of erythemal irradiance (Jarosławski et al., 2003). This study also showed no wavelength dependence of AOD in the UVB/UVA (310–340 nm) range and even negative AE in the 310–320 nm range (see Fig. 6 of Jarosławski et al., 2003). In TUV simulations of the erythemal irradiance we use fixed other aerosols characteristics derived as the 2004–2023 climatological means based on the measured the Cimel Sun photometer values at 440 nm. Raptis et al. (2018) pointed out that variations in the single scattering albedo (SSA) in the UV range could be omitted in rural, pristine areas with low AOD values. Belsk appears to belong to this category, with a mean $AOD_{340\ nm}$ of approximately 0.3. TUV simulations of clear-sky noon erythemal irradiance, assuming an $AOD_{340\ nm}$ of 0.3 and an AE of 0, showed that changes in SSA from 0.82 to 0.96 (i.e. the minimum and maximum SSA averaged values for the 2004–2023 spring–summer seasons at Belsk) yielded variations in the noon erythemal irradiance of only about ±5% relative to the value obtained using the mean SSA value of 0.92 for this season. Similarly, the effects of clouds on UV were parameterised using a limited number of proxies: the strength of attenuation of global solar radiation by clouds, as indicated by CI, SunDur and G. In addition, a constant surface albedo of 0.03 was used throughout the year. It cannot be excluded that many other characteristics of clouds influence surface radiance, e.g. liquid water content, optical depth, specific cloud types, their bases and tops, and multiple light scattering between clouds at different levels and between snowy ground and clouds. Long time series of such parameters were not available to us. However, the choice of aerosol and cloud proxies used in erythemal UV modelling ensures excellent agreement between the modelled and observed annual erythemal radiation (ERE), as shown in Fig. 6, where the smoothed modelled ($GPR_{all}$, blue solid curve) and observed (black solid curve) time series are almost superimposed. The descriptive statistics of the relative differences between the model and observations are −0.04%, 0.89%, 0.98 and 1.25% for mean, mean absolute value, correlation coefficient and standard deviation, respectively. These values also confirm the effectiveness of using a limited number of proxies in analyses of long-term variability in surface UV radiation. Small differences in the long-term patterns of the annual ERE with the model using a full set explaining variables ($GPR_{all}$) and model ($GPR_{TCO3*\ \&\ SunDur\ \&G}$) using satellite $TCO_3$ and typical characteristics of clouds (sunshine duration, global solar irradiance), which are measured at numerous weather stations, suggest that it will be possible to reconstruct and trust in the annual ERE over a larger area surrounding Belsk.

The strength coefficient ($SC\_MOD_X$) of the selected X driver effect on the annual ERE obtained with each of the models considered here (i.e. the TUV model, the advanced and simplified GPR model, and the SLR model) is proposed to establish a ranking of the drivers. A strong dominance of cloud driver appeared for GPR models as the ratios between $SC\_MOD_X$ values are in the range of 1.5−2.5. This dominance is not so apparent for SLR models, where the ratios are ~1 in the range of 0.85−1.64

(Table 9). In the case of the TUV model, the effects of $TCO_3$ and aerosols are almost equal for the whole UVR observation period and for the period 1976-1999, as the ratio between the respective strength coefficients is ~ 1. The dominance of the $TCO_3$ effect was found in the period 2000-2023 with a ratio of 1.43, which corresponds to the cleaning of the atmosphere over Belsk in connection with a European-wide policy to improve air quality.

A comparison of linear and non-linear superposition of the effects induced by clear sky and cloud proxies shows the importance of non-linear interactions between these proxies in producing the long-term changes of ERE. A measure, the interaction strength coefficient (ISC), is proposed to assess the strength of such interactions. This is defined as the square root of the mean square of the difference between the linear and non-linear model values, represented by the red circles and blue crosses in Fig. 6 and Fig. 7. For these Figures, small ISC values of 1.53% and 0.84%, respectively, are calculated, which means that the interaction effect between clear sky and cloud proxies can be neglected in trend analyses, which is also supported by a close similarity in the smoothed ERE patterns with the linear and non-linear model. It is also found that the non-linear interaction between the effects of $TCO_3$ and $AOD_{340\,nm}$ on ERE for cloudless conditions gives ISC of 0.13%, meaning that the simple linear model (2) can replace the time-consuming simulations of the TUV model (Fig. 5).

## 6 Conclusions

This paper proposes a method to reveal the sources of long-term variability and the strength of the induced UVR variability due to individual forcing factors. The linear and non-linear superposition of clear sky and cloud effects are also calculated and compared with each other and with observations. The annual erythemal exposure in Belsk in the last decade of the 1976−2023 measurements is about 15% higher than in the first decade. Monthly erythemal exposure during the warm season (April−September) shows a similar increase. The positive UVR trend in the 1980s and 1990s was a specific superposition of stratospheric ozone, aerosol and cloud effects. However, from a public health perspective, the small but statistically significant upward trend in seasonal (June−August) UVR exposure at Belsk since about 2000 seems to be of particular interest, as people spend more time outdoors during these summer months. At present, it is difficult to say whether such a trend is due to local effects or whether it is a signal of an emerging non-local, large-scale UVR forcing, probably related to climate change, with a greater frequency of days with cloudless skies. A method is proposed to build a set of models (such as the non-linear GPR models in Section 4.2.1) to investigate sources of long-term variability in ERE, with indices to quantify the influence of selected factors on UVR and the strength of interactions between them. These indices can be applied to any model and set of variables explaining ERE variability.

**Author Contribution.** Conceptualisation, AC and JK; methodology, AC, JK, JJ, PS, and AP; validation, AC and JK; visualisation, AC; writing (original draft preparation), AC and JK; writing (review and editing), AC, JK, and PS; funding acquisition, JJ and AP. All authors have read and agreed to the published version of the paper.

**Competing interests.** The contact author has declared that none of the authors has any competing interests.

**Acknowledgements.** This work was partially financed by the Chief Inspectorate of Environment Protection, contract number GIOŚ/31/2023/DMŚ/NFOŚ. The authors acknowledge ACTRIS research infrastructure for providing aerosol data. Operation of the Polish part of the aforementioned research infrastructure has received financial support from the Minister of Science, as outlined in agreement no. 2024/WK/04. The authors would also like to thank the DeepL tool (https://www.deepl.com/) for its help in improving the English language.

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
