# Peer review of "Trends in the erythemal radiant exposure from re-evaluated measurements (1976–2023) with biometers at Belsk, Poland, and their sources from corresponding ozone, aerosol, and cloud observations"

_EGUsphere, 2025_

## Author Comment (AC1)

**Response to the Reviewer 1**
(*reviewer text is in italic*)

**General comments**

*The authors analyze a very long (the world's longest as they claim) homogenized series of measured erythemal solar irradiance. Then, for the same period (1976 - 2023) they reconstruct the series using different proxies to determine the contribution of different factors to the observed trends. Although the manuscript is well structured and well written, there are some major issues that must be considered by the authors prior to the submission of a revised version of the paper.*

*To homogenize the series the authors have scaled the measurements using the reconstructed series of the noon erythemal doses. In my opinion, the scaling should be performed using low-turbidity days to avoid biases due to the impact of changes in the prevailing aerosol species. Gradual changes in aerosol composition would affect optical properties such as the single scattering albedo (SSA) and the Angstrom Exponent (AE), that can affect UV doses and their changes have not been taken into account. Furthermore, the characteristics of different sensors possibly affect the results, even after the scaling to the modelled doses. For example, imperfect angular response, or temperature dependence are possible different for different sensors. If such effects have not been considered they increase the uncertainty in the homogenized series. There should be at least some discussion regarding the remaining uncertainties after the homogenization.*

The discussion has been added and in the revised manuscript we have:

"The uncertainty of the homogenisation procedure can be assessed by comparing the descriptive statistics of the differences between the regression model and the observed daily EREs from the KZ meter (operated at Belsk from 2014 to the present day with no required adjustments, as evidenced by the excellent agreement with co-located Brewer spectrophotometer observations; Krzyścin et al., 2025) with the corresponding statistics of the model-observation differences using the homogenised RB and SL501A measurements (1976–1992 and 1993–2013, respectively). The model was trained using data from 2014 to 2023 and applied to pre-2014 periods. For full-year data, the mean relative difference between the model and the KZ observations is 1.4%, whereas it is between −2.7% and 1.6% for the RB and SL501A meters, respectively. The respective standard deviations were found to be 10.4%, 13%, and 14.6% (see Table 4 in Krzyścin et al., 2025). These descriptive statistics are only a few percentage points higher than those obtained for the KZ measurements. Thus, the homogenisation procedure for the period 1976–2013 resulted in a slight increase in uncertainty due to differences in the individual instrumental characteristics of the RB and SL501A meters, as well as the limited aerosol input data (annual mean AOD) used in the regression model, which was derived from sparse Sonntag pyrheliometer measurements in that period." (L. 436-447).

Given the above, considering the specific characteristics of aerosols will not change the accuracy, which is already within the range of measurement error.

*The proxies that have been used for the series reconstruction do not always correspond to UV-B (i.e., the part of the solar spectrum that mostly contributes to the erythemal doses). For example, the relationship between the AOD at 340 nm and the AOD below 315 nm depends on the AE (practically from the aerosol species) which has been assumed to be equal to 0. The SSA has not been considered and can also have significant impact on the UV trends. The effects of clouds depend on their type and properties. There should be at least some discussion on the uncertainties related to these factors.*

The text has been added to the revised manuscript to explain this problem:

"At first glance, the proposed indices to explain changes in annual and seasonal (summer) erythema exposure appear to be only loosely related to the UV radiation attenuation in the atmosphere. To account for the effect of aerosols, the daily averaged $AOD_{340\ nm}$ was used with an Ångström exponent of 0 across the UV range, while other features were held constant. SunDur, G and CI (i.e. parameters related to global solar radiation rather than UV radiation) were chosen to account for the influence of clouds. In addition, a constant surface albedo of 0.03 was used throughout the year. However, this choice of UV indices ensured excellent agreement between modelled and observed annual ERE, as shown in Fig.6 where the smoothed modelled ($GPR_{all}$, blue solid curve) and observed (black solid curve) curves were almost superimposed on each other. The descriptive statistics of the relative differences between the model and observations are −0.04%, 0.89%, 0.98 and 1.25% for mean, mean absolute value, correlation coefficient and standard deviation, respectively. These values also confirm the effectiveness of using a limited number of proxies in analyses of long-term variability in surface UV radiation."( L. 465-474).

Specific comments are provided below.

**Specific comments**

*L8: "measured erythemal " instead of "erythemal". There are reconstructed series that are longer.*

*L10: "observations period" instead of "observations"*

The proposed changes were included in the revised manuscript.

*Introduction: In additions to being the main source of vitamin D, exposure to UVR has other positive effects. E.g.,:*

*https://www.tandfonline.com/doi/full/10.4161/derm.20013*

*https://www.mdpi.com/1660-4601/13/10/1028*

The following sentence was added to the introduction (L. 34-36): "In additions to being the main source of vitamin D, exposure to UVR has other positive effects. E.g.,: lowering the blood pressure, psoriasis clearing, improving mood by endorphin release, and increases the melanin production in skin (Juzeniene and Moan, 2012; Trummer et al., 2016)."

*Section 2.2: How was the clearness index "translated" to clearness index for ERE? Please provide more details. Furthermore, what about other aerosol properties (e.g., SSA, AE) that affect ERE? Furthermore, please provide a reference for the erythema actions spectrum that has been used to calculate ERE.*

The following details have been provided in the revised manuscript:

"The CI value, which is calculated on the basis of daily ERE, $CI_{ERE}$, is proportional to the CI value from global solar radiation. The conversion formula is often in the form: $CI_{ERE} = \alpha CI^{\beta}$, where the coefficients are derived empirically and may depend on solar zenith angle, resulting in a smaller $CI_{ERE}/CI$ ratio at lower solar elevation (e.g. Krzyścin et al. 2025). " (L. 113-115)

It seems that SSA and AE affect only slightly the $CI_{ERE}$, as it is calculated as the ratio of the all-sky to clear sky ERE. This means that, in cases where variations in SSA and AE are a concern (i.e. for clear or almost clear conditions), $CI_{ERE}$ is, by definition, close to 1.

ERE was measured by the broadband meters, and instrumental action spectra were close to the actual (at that time) erythemal action spectrum provided by CIE (Commission Internationale de l'Eclairage). All of the differences between spectra were taken into consideration in the homogenisation procedure. The action spectrum for erythemal weighting for the Brewer spectral measurements was CIE 2019 (which is the same as CIE 1998). The reference to action spectrum has been given in line 57.

*L184-186: Given that the AE can practically range from ~ 0 (e.g., for dust) to ~2 (e.g., for biomass burning aerosols), and that the greatest contribution to ERE comes from wavelengths at 306 – 308 nm, there can be a difference of up to ~20% between AOD at 340 nm and the AOD at wavelengths that contribute more significantly to ERE. There should be at least some discussion about that. Furthermore, during the cold period, can changes in surface albedo have played a role? Does the assumption of a default surface albedo introduce any uncertainty?*

We agree that the characteristics of aerosols in the UV-B range are important for analysing surface erythemal irradiance. However, as discussed in the authors' response to the second General Comment, knowing the detailed characteristics of the aerosols for the study of long-term variability in annual and summer erythemal ERE is much less important.

The effects of albedo can be inferred from the selected proxies using a machine learning approach when a much higher G value is met under cloudless conditions in the cold period.

*L233: "monitored" instead of "monitoring"*

It was changed in the manuscript.

*L277-278: What does 3%p means?*

per cent point (%p) is defined in line 298.

*L387: Delete "much"*

It was changed in the manuscript.

---

## Author Comment (AC2)

**Response to the Reviewer 2**
(*reviewer text is in italic*)

*The paper 'Trends in the erythemal radiant exposure from re-evaluated measurements (1976−2023) with biometers at Belsk, Poland, and their sources from corresponding ozone, aerosol, and cloud observations ; by Czerwinska et al., investigates a time series of erythemal solar irradiance 1976-2023 in Belsk, Poland and analyses trends and potential factors influencing it. This topic is very relevant for a better understanding of the evolution of solar uv radiation received at the surface, in particular if there is an increasing or decreasing trend and which factors possibly influence this evolution. The strength of the paper is that the influencing factors like total ozone, aerosol load, cloud coverage are taken into account by several statistical means and time scales. The manuscript is fitting into the scope of Atmospheric Chemistry and Physics. The manuscript is well written and I can recommend publication for ACP after some revision, described below.*

*General comments:*

*The authors claim that their time series of homogenized erythemal uv dose is the longest globally. This should be substantiated by giving at least some references of other relevant long time series and respective references.*

Some of the time series are listed and referred to in the article in ESSD (Krzyścin et al., 2025). As these two articles come together, only the brief statement was added to the text in (L. 58-60):

"Time series of UV ground-based measurements longer than four decades are very rare. They are not erythemally weighted (Chubarova et al., 2000) or do not come from biometers, but rather spectral measurements (NDACC, 2025, WOUDC, 2025)."

*The authors should give for their instruments used for deriving the Erythemal Radiant Exposure (ERE) the wavelength range measured, mentioning also 'broadband' (or spectral in case), this could e.g. be integrated in table 1. It should be indicated if these sensors used by the authors have been calibrated regularly against reference standards.*

All of the ERE meters used in this study were erythemal broadband meters. The only exception is the Brewer Spectrophotometer #64, which was used for regular calibration since 2013, but the data from this instrument was not used in this paper. Thus, all ERE measurements come from broadband meters in this study. Wavelength ranges were added in the Table 1. The detailed information about biometers' calibration can be found in the co-article in ESSD (Krzyścin at al., 2025).

*The authors explain (also via the co-article in ESSD) that the quality of the ERE time series of the KZ sensors was accessed by checking with ERE observations of the Brewer #64. The Brewer has however another uv range measured. How was the ERE therefore derived in order to assure comparability? Further, for the Brewer#64: in section 2.2, Ancillary data, it is not included in the TCO3 database? Why?*

The following explanation has been added in the text (L. 55-58):

"The SHICRivm algorithm is used for quality checks and to extend the Brewer spectra to 400 nm (Slaper et al, 1995). To retrieve daily erythemal radiant exposures for comparison with KZ, spectra are weighted with the CIE erythemal action spectrum (CIE 2019) and integrated over wavelengths (from 280 to 400 nm) and time (sunrise to sunset)."

We did not use the TCO3 from Brewer measurements, as the data is not available since the beginning of ERE measurements. Furthermore, the Dobson measurements have been quality checked since the beginning of the TCO3 observations in 1963. The data from Brewer has many loopholes and overlaps with Dobson, thus it was not included in this database.

*The introduction should also include a WMO definition of the erythemal (uv) irradiance and its relation to the commonly communicated uv index.*

The following text has been added in the text (L. 40-42):

"Erythemal UV irradiance is the intensity of UV radiation that causes sunburn, i.e., UV spectrum weighted with action spectrum of erythema (sunburn) appearance and integrated over the wavelength range from 280 to 400 nm. The commonly used 1 UV Index is equivalent to 25 mW m$^{-2}$ of erythemal UV irradiance (WHO, 2002)."

**Specific comments:**

*Line 50: the Brewer #64 spectrophotometer is mentioned here, but not a bit further above in the list of instruments used for monitoring important parameters; The Brewer should also be listed there briefly (since when, where?). Or was its data not used any further?*

The data was used only for checking the quality of KZ meter's measurements (in this study), thus the Brewer is mentioned only briefly in the introduction.

*Lines 76-77: 'the months 1985 (June and July) were exluded' …: what does this mean? Only for the year 1985, or for all years these two months? If June and July have been excluded for all years, then it would be necessary to do the same calculations once with including them in order to check the uncertainty budget. If only 1985, then why this year?*

It was only in 1985 because there was a lack of data in June and July. Thus, it was not possible to count the monthly erythemal radiant exposure for those months, and filling the gap with the long-term mean for the months that contribute the most to the yearly mean seems to be inappropriate. The sentence was corrected to make it more clear (L. 92-94): "The months of 1985 (June and July) were excluded from the analyses because of the lack of data, and replacing them with long-term monthly means seems to be incorrect, as UVR in these months contributes mostly to the annual ERE."

*Section 2.2 Ancillary Data: It should be more detailed how the time series for these individual influencing factors have been assured to be homogeneous; e.g. the TCO3 between Dobson and satellite observations, and the G_o values (if ECMWF or from MERRA-2)? How was it decided to take the one or the other? In case of AOD from Aeronet – which data level has been taken? SunDur: 'percentage of daily duration': daily duration means from sunrise to sunset, or fixed day duration?*

As for TCO3, in ESSD paper (Krzyścin et al. 2025) we explained in more details the code flags for TCO3 and how the series was constructed. Dobson results represent more than 90% of the time series. Satellite measurements are in good agreement with Dobson measurements, i.e., the mean relative difference is less than 1%. As for G0, we compared measured G for cloudless sky conditions at Belsk with G0 from MERRA-2 and ERA5. We found that the best fit was for the mean values of MERRA-2 and ERA5. Before 1980, we used ERA5, but the data was corrected with the mean bias. In case of AOD, we used data level 2.0. Daily duration means from sunrise to sunset. These details were added in the paper in lines: 100-101, 103-104, 106, 109-111.

*Lines 101-105: The formulation why to take years 1999/2000 as turning point can be improved. Later on, the authors give more details (lines 135-139), but at this place here, it is a bit confusing (especially the formuation 'somewhere in between'). When looking at the values in table 3, the R2 value for year 2000 is always lower (in one case equal) than the R2max for another year – although in line 137 it is stated to choose the year with R2 max. It feels thus that the choice for years 1999/2000 needs an improved motivation. E.g. volcano eruptions (Pinatubo, …), enter into force of the Montreal Protocol, instrumental issues, further statistical change point analysis….*

Values of R2 for the months with statistically significant trends, are not lower for R2000 of more than 0.05 than for R2max, and for yearly values the difference is only 0.01, while for seasonal, the R2 values are the same. As the differences are so small, we chose the year 2000 as the turning point, to choose the same turning point for all time series, while the Tmax would be different for each time series. The statement in line 137 (now line 161) is not correct and was improved. The word "selected" was changed to "calculated". We also changed the previous fragment to: "Our first guess was that a linear trend for the whole period 1976−2023 would not be applicable because of a trend shift in the annual ERE, visible in the smoothed data, that occurred between the years 1995−2015. The year 2000 was chosen as the trend-turning year that divided the whole observing period into two sub-periods (1976−1999 and 2000−2023) with equal length, and also to choose the same turning point for each time series. The effect of this selection on the trend values is discussed further in this Section." (L. 125-129)

*Lines 184-188: In addition to the comments of the other reviewer: How valid is the assumption that AOD at 340 nm is representative for the UV-B range? There should be some discussion on it, including some references to measurements of AOD in the UV-B.*

The discussion was added (L. 465-474): "At first glance, the proposed indices to explain changes in annual and seasonal (summer) erythema exposure appear to be only loosely related to the UV radiation attenuation in the atmosphere. To account for the effect of aerosols, the daily averaged AOD340 nm was used with an Ångström exponent of 0 across the UV range, while other features were held constant. SunDur, G and CI (i.e. parameters related to global solar radiation rather than UV radiation) were chosen to account for the influence of clouds. In addition, a constant surface albedo of 0.03 was used throughout the year. However, this choice of UV indices ensured excellent agreement between modelled and observed annual ERE, as shown in Fig.6 where the smoothed modelled (GPRall, blue solid curve) and observed (black solid curve) curves were almost superimposed on each other. The descriptive statistics of the relative differences between the model and observations are −0.04%, 0.89%, 0.98% and 1.25% for mean, mean absolute value, correlation coefficient and standard

deviation, respectively. These values also confirm the effectiveness of using a limited number of proxies in analyses of long-term variability in surface UV radiation."

*Line 360, table 9: Would it be possible to increase readability? It is not intuitive to read connected numbers so easily*

The table was changed to increase readability.

*Line 379: 'neighbouring stations: do the authors mean Hradec Kralove and Lindenberg? Then this should be clarified ('these neighbouring …')*

Yes, it was clarified.

*Lines 383-385: For the mentioned numbers for Reading and Sodankylä – please clarify to which reference these belong*

The reference is given in the previous sentence, i.e. Fountoulakis et al. 2020 (https://doi.org/10.3390/environments7010001). "Fountoulakis et al. (2020) found that local effects mainly related to the aerosols, albedo, and cloud changes in Europe, were important in shaping the UVR trend. In the case of Reading (51.4° N, 0.9° W), UK, and Sodankylä (67.4° N, 26.6° E), Finland, downward trends in the 307.5nm irradiance were even found between 1996 and 2017."

*Discussion section on uv trends: Are there references for trend analyses reaching further than the ones mentioned? E.g.: https://doi.org/10.5194/acp-22-12827-2022; https://doi.org/10.5194/acp-21-18689-2021; https://doi.org/10.1007/s43630-024-00658-8; https://doi.org/10.1002/joc.7803*

Most of the proposed references were added into the discussion (L. 430-435):

"This conclusion aligns with other studies. Fountoulakis et al. (2021) found that in 2006-2020, the levels of UV irradiance at Aosta, Rome and Lampedusa (Italy) increased mainly due to changes in cloud clouds or/and aerosol. Lorenz et al. (2024) also attribute the increasing trend in erythemal UV radiation in Dortmund (Germany) in 1997-2022 to a change in cloud cover as the primary driver. In the reconstructed series in Novi Sad (Serbia) in 1971-2018, Malinović-Milićević et al. (2022) found that there were no significant trends in the last two decades, but there was an increase in the number of days with high daily radiant exposures. The authors also stated that the cloud cover had more influence than TCO3."

*Lines 402-411: this paragraph fits rather to the conclusions, as it describes the results rather than to discuss them*

The part of the paragraph was moved to the conclusions (L. 480-482), but most of it, in the authors' opinion, fits better in the discussion because of too many details. This conclusion is further discussed with other studies, suggested by the Reviewer (L. 424-435).

*Lines 418-419: Is there a source reference for the statement that the atmosphere over Belsk was getting cleaner?*

Yes, Posyniak et al., 2016 (https://doi.org/10.1515/acgeo-2016-0026)

*Line 423: 'red circles and blue crosses': these are not the physical entities behind; please write out the meant individual factors/proxies.*

The sentence was changed to: "This is defined as the square root of the mean square of the difference between the linear and non-linear model values, represented by the red circles and blue crosses in Fig. 6 and Fig. 7." (L. 458-460)

*Lines 435-437: why should there no further increase of UVR be expectable? Total ozone is highly variable, also atmospheric aerosol load and how these factors will change is not determined, in particular taking climate change into account. Climate change is also expected to impact on cloud cover. Overall warmer temperatures and a purer atmosphere support less cloud cover.*

The authors agree with the Reviewer and changed the sentence in the conclusions to: "The positive UVR trend in the 1980s and 1990s was a specific superposition of stratospheric ozone, aerosol and cloud effects." (L.484-485)

***Technical comments:***

*Line 88 and in figure 1 caption: define CI for the clearness index*

*Line 126: please change to 'The largest differences ...'*

*Line 282: '... in the per cent ...' please skip the 'the'*

*Line 288ff: 'ozone & aerosols'; better write 'combined ozone and aerosol impact' or similar*

*Line 324: in brackets: (approximately between 0.8 to 0.9)*

*Line 325: skip 'per cent' (% already there)*

*Line 378: please change to 'Republic and Lindenberg'*

*Line 383: please rephrase: 'mainly related to the changes in aerosol load, albedo and cloud cover ...'*

All the changes from technical comments were applied.

---

## Author Response (AR2)

**Response to the Editor decision: Reconsider after major revisions:**

*The authors have considered the reviewer comments. However one of the major comments addressed by one reviewer was the inclusion (or non-inclusion of AE and SSA changes and also cloud type related discussion).*

*More details on AE(=0) choice that seems not so reasonable, consideration of SSA changes and also some basic discussion on different cloudiness and types will provide more evidence on the validity of the method and the whole analysis."*

The discussion of these problems was added to the new version of the revised manuscript:

"To account for the effects of aerosols, the daily-averaged AOD at 340 nm was used with an Ångström exponent (AE) of 0 across the UV range, while the other features were held constant. This approach is consistent with the homogenisation procedure for Belsk's UV measurements from 1976 to 2023, as proposed by Krzyścin et al. (2025). Our previous study examined Brewer and the Cimel Sun photometer measurements at Belsk and showed that AE was close to zero in modelling of erythemal irradiance (Jarosławski et al., 2003). This study also showed no wavelength dependence of AOD in the UVB/UVA (310–340 nm) range and even negative AE in the 310–320 nm range (see Fig. 6 of Jarosławski et al., 2003). In TUV simulations of the erythemal irradiance we use fixed other aerosols characteristics derived as the 2004–2023 climatological means based on the measured the Cimel Sun photometer values at 440 nm. Raptis et al. (2018) pointed out that variations in the single scattering albedo (SSA) in the UV range could be omitted in rural, pristine areas with low AOD values. Belsk appears to belong to this category, with a mean $AOD_{340\,nm}$ of approximately 0.3. TUV simulations of clear-sky noon erythemal irradiance, assuming an $AOD_{340\,nm}$ of 0.3 and an AE of 0, showed that changes in SSA from 0.82 to 0.96 (i.e. the minimum and maximum SSA averaged values for the 2004–2023 spring–summer seasons at Belsk) yielded variations in the noon erythemal irradiance of only about ±5% relative to the value obtained using the mean SSA value of 0.92 for this season.

Similarly, the effects of clouds on UV were parameterised using a limited number of proxies: the strength of attenuation of global solar radiation by clouds, as indicated by CI, SunDur and G. In addition, a constant surface albedo of 0.03 was used throughout the year. It cannot be excluded that many other characteristics of clouds influence surface radiance, e.g. liquid water content, optical depth, specific cloud types, their bases and tops, and multiple light scattering between clouds at different levels and between snowy ground and clouds. Long time series of such parameters were not available to us. However, the choice of aerosol and cloud proxies used in erythemal UV modelling ensures excellent agreement between the modelled and observed annual erythemal radiation (ERE), as shown in Fig. 6, where the smoothed modelled ($GPR_{all}$, blue solid curve) and observed (black solid curve) time series are almost superimposed."  L. 448-468.

The structure of the discussion was changed to ensure that the section on uncertainty remained consistent.

The following references were added to the reference list:

Jarosławski, J., Krzyścin, J.W., Puchalski, S., and Sobolewski, P.: On the optical thickness in the UV range: Analysis of the ground-based data taken at Belsk, Poland, J. Geophys. Res., 108(D23), 4722, https://doi.org/10.1029/2003JD003571, 2003.

Raptis, P., Kazadzis, S., Eleftheratos, K., Amiridis, V., and Fountoulakis, I.: Single Scattering Albedo's Spectral Dependence Effect on UV Irradiance, Atmosphere, 9, 364, https://doi.org/10.3390/atmos9090364, 2018.